# A Distinct Anti-EBV DNase Profile in Patients with Undifferentiated Nasopharyngeal Carcinoma Compared to Classical Antigens

**DOI:** 10.3390/v15112158

**Published:** 2023-10-26

**Authors:** Hamid Melouli, Abdelhalim Khenchouche, Fouzia Taibi-Zidouni, Dahmani Salma, Nassim Aoudia, Djamel Djennaoui, Tewfik Sahraoui, Samir Benyahia, Fatima Zohra El Kebir

**Affiliations:** 1Viral Oncogenesis Laboratory, Pasteur Institute of Algeria, Algiers 16000, Algeria; hmeloulifzh@gmail.com (H.M.);; 2Laboratory of Applied Biochemistry, Ferhat Abbas, Setif 1 University, Setif 19000, Algeria; 3Otorhinolaryngology Department, Mustapha Pacha Hospital, Algiers 16000, Algeria; 4Laboratory of Developmental Biology and Differentiation, Es-Sénia University, Oran 31000, Algeria

**Keywords:** undifferentiated nasopharyngeal carcinoma, diagnosis, EBV, VCA, EA, DNase

## Abstract

Nasopharyngeal cancer (NPC) is a prevalent type of cancer that often takes the form of undifferentiated carcinoma in the Maghreb region. It affects people of all ages. NPC diagnosis, mainly based on detecting Epstein-Barr virus (EBV), has not been well evaluated in North Africa. We compared the classical EBV serological tests using indirect immunofluorescence to the detection of EBV DNase antibodies by immunoblot in Algerian NPC patients. Significant variations were observed among different age groups of patients regarding the presence of VCA-IgA antibodies (0–14 and ≥30 years old, *p* < 0.0001; 15–19 and ≥30 years old, *p* < 0.01) and EA-IgA (0–14 and ≥30 years old, *p* < 0.01; 15–29 and ≥30 years old, *p* < 0.05). Differences were also noted in the titers of IgA anti-VCA and anti-EA antibodies across the three age groups. Some patients under the age of 30 with detectable IgG anti-VCA antibodies had undetectable IgA anti-VCA antibodies. These patients had a strong anti-DNase IgA response. However, older individuals had a higher level of anti-DNase IgG. Before treatment, children had strong DNase reactivity as indicated by specific IgA antibodies. Young adults had high IgA anti-DNase response, but the elderly (90.9%) had a lower response for these antibodies. Following therapy, the children retained high levels of IgA anti-DNase antibodies, and 66% of the young adults demonstrated robust antibody reactivity against DNase. In contrast, IgG responses to anti-DNase were low in children. This study demonstrated the utility of anti-DNase responses in the diagnosis and prognosis of NPC.

## 1. Introduction

Epstein-Barr virus (EBV), a lymphotropic herpesvirus, is known for causing infectious mononucleosis and malignant lymphomas. This virus is also strongly associated with undifferentiated nasopharyngeal carcinoma (UCNT), a common tumor type in Algeria that affects children, young adults, and adults. In contrast, just the adult peak is seen in Asia. The diagnosis of UCNT often relies on the detection of IgA antibodies against EBV viral capsid antigen (VCA) and early antigen (EA), although molecular techniques are recognized for their enhanced sensitivity [1]. VCA-IgA antibodies are uncommon in healthy people, giving them a dual function in the diagnosis and prognosis of UCNT [2,3,4]. However, a significant number of North African individuals under the age of thirty have undetectable serum VCA-IgA levels [5,6,7]. This suggests a congenital IgA deficiency, occurring in approximately 1 out of every 700 Caucasians [8]. Several other factors may contribute to this phenomenon, including recent EBV infections in the region, the diversity of EBV strains prevalent among younger individuals in North Africa, or the influence of individual genetic factors that can impact an individual’s immune response to infections.

Advancements in EBV biomarkers have been pursued to enhance the diagnosis of nasopharyngeal cancer (NPC) [9,10]. The specificity and sensitivity of EBV testing using single markers have been debated [11,12]. Some EBV markers were proposed to improve the diagnosis and prognosis of NPC. Particularly, the BARF1 protein has been identified in the serum and saliva of North African NPC patients, with lower levels of anti-BARF1 antibodies observed in these patients [13]. The quantification of circulating EBV DNA has emerged as a method with superior accuracy in diagnosing and monitoring NPC [14]. However, establishing a consensus on the lowest limit of plasma EBV detection remains a challenge [15,16]. Additionally, the BGLF5 gene product (DNase) has shown promise as a robust marker. NPC sera contained high levels of antibodies against DNase, with fluctuations during periods of remission and relapse [17,18,19]. The baculovirus expression system has been used for the expression of recombinant DNase within insect cells [20]. The product of this expression system has shown increased IgA synthesis in NPC patients, as demonstrated by Chang et al. [21].

Our study highlighted the limitations of the immunofluorescence assay (IFA) using viral capsid antigen (VCA) and early antigen (EA) in patients under 30 years of age. We used the immunoblot enhanced chemiluminescence (ECL) assay to demonstrate its ability to detect antibodies against DNase in all patients. This test’s remarkable sensitivity and specificity make it helpful for providing significant assistance and serving as a complementary tool for an accurate diagnosis.

## 2. Materials and Methods

### 2.1. Patients and Controls

Between 2012 and 2021, a total of 622 cases of undifferentiated nasopharyngeal cancer (UCNT) were classified histologically as NPC tumors according to the World Health Organization classification [22] and staged according to the 7th Edition American Joint Committee on Cancer/Union for International Cancer Control (AJCC/UICC) [23]. The average age was 31.17 years in patients (SD ± 17.4), with extremes of age from 10 to 77 years. From this pool, 156 patients diagnosed at the Otolaryngology Department of the Mustapha Pacha Hospital in Algiers, Algeria, were randomly selected. Blood samples were taken before receiving the treatment. After treatment, other blood samples were collected from 76 patients (Table 1). Patients were sampled under free and informed consent, following the Declaration of Helsinki. As part of the negative control group, serum samples were gathered from 112 disease-free individuals. These samples were sourced from a national sero-epidemiological study conducted to assess immunity to poliovirus among children and adolescents. Additionally, some samples were obtained from blood transfusion centers within Algiers hospitals.

### 2.2. Cell Lines and DNase

Dr. Ooka of the Molecular Virology Laboratory at RTH Laennec in Lyon, France, provided us with the recombinant EBV DNase as part of two Algerian inter-university projects: French 95MDU319 (CMEP) and 05MDU663 (CMEP). The recombinant EBV DNase of 55 KDa (ORF BGLF5: 12907–122451 positions) was obtained from *Spodoptera frugiperda* 9 cells infected with recombinant baculovirus (AcNPV37: cDNA37 (1.4 Kb)) corresponding to P3HR-1 incorporated within the baculovirus genome region encoding polyhedrin and purified using phosphocellulose and cellulose DNA columns.

### 2.3. Immunoenzymatic Tests

Serological tests for the detection of anti-VCA and anti-EA antibodies were performed using commercial immunofluorescence kits (IFA system, Euroimmun AG, Luebeck, Germany). The sera samples were serially diluted by a factor of 2, starting from 1:10 dilution. Serum endpoint titers of antibody to VCA and EA equal to or greater than 1:10 were considered positive.

### 2.4. Immunonephelometry

The quantification of total serum IgA immunoglobulin was carried out using a Nephys-Analyzer Medical System scattering 660 nm light. Serum samples were pre-diluted at 1:20 before analysis. The IgA concentration in unknown samples was derived from a calibration curve that used 6 calibrators at different levels (IgA high control ref: NA-CPPA; IgA medium control ref: NA-CPPM; IgA low control ref: CPPB) and a standard of known concentration (IgA: 0.9–4.5 g/L). If the result was higher than the upper limit of the measuring range another dilution of ¼ was performed, and if the result was lower than the lower limit of the measuring range, the test was performed without dilution.

### 2.5. SDS-PAGE and ECL-Immunoblot

A sample of 5 µg of DNase was subjected to 10% acrylamide/biscrylamide gel electrophoresis under denaturing conditions in Tris-glycine buffer, pH: 8.3 (Tris 0.4 M, glycine 0.4 M) after being heated at 100 °C for 5 min in denaturation buffer (Tris 0.5 M pH: 6.8, glycerol 50%, SDS 20%, 2 β-mercaptoethanol 5%, bromophenol blue 0.5%). The nitrocellulose membrane (type optician BA-S 85, GE Healthcare, France) was saturated with a solution of 5% non-skimmed milk in phosphate-buffered saline with 0.5% Tween 20 (PBST). It was then incubated for 2 h with 200 μL of patient sera diluted at 1:10 in 1% PBST–milk at room temperature. The membrane was in contact with peroxidase-conjugated goat anti-human IgG or IgA antibodies (Sigma Immunochemical) diluted 1:1000 in 1% PBST–milk for 1 h at room temperature. The ECL kit (GE Healthcare) identified the anti-DNase antibodies. As controls, a negative serum (BA003/1) from an uninfected European individual and a positive serum (TU115: VCA IgA = 5120 and EA IgA = 2560) from a Tunisian patient with NPC were used. The colored markers (Rainbow RPN 756, GE Healthcare) were treated similarly. The presence of a 55 KDa band indicating the product of the primary EBV BGLF5 reading frame was characterized as reactivity.

Each sample lane was scanned and pixel measured using ImageJ software (v. 1.8.0, Bethesda, MD, USA) for normalized quantification of the Western blot. Diluted TU115 (1/10) was given a value of 1. After lane background subtraction, the relative pixel value of each sample was obtained using the following equation to obtain a semi-quantitative band value:$$RPV\;(relative\;pixel\;value)=\frac{NCP\;sample\;pixels-lane\;background\;pixels}{TU115\;pixels-background\;pixels}$$

### 2.6. Statistical Analysis

The Khi2 test was used to calculate the difference in the prevalence of anti-VCA and anti-EA antibodies between patients with NPC and the control groups. The Wilcoxon–Mann–Whitney test was used to evaluate the distribution of anti-VCA and anti-EA IgA titers in groups of NPC patients and healthy control subjects. The Kruskal–Wallis test was used to determine the difference in titer distribution by age and clinical stage. All statistical analyses were carried out with the Epi Info program, version 6. A significant difference was defined as a *p*-value less than 0.05.

## 3. Results

The distribution of cases based on 5-year age groups revealed the occurrence of two distinct age peaks (Figure 1); the first peak happening between the ages of 15 and 19, followed by the second peak occurring between the ages of 50 and 54. Among those diagnosed, 33.08% were identified before reaching the age of 30, and 9.51% were diagnosed before the age of 15. The gender distribution of cases followed the traditional pattern: approximately equal for both sexes before the age of 20, followed by an increase beyond the age of 30, resulting in a male–female ratio of 3.71. Comparable findings from other sources show that the incidence rate in men is two to three times that of women [24].

We divided the patients into three groups based on their age to better discern the differences in immunological responses to EBV markers and the effect of age on the performance of the EBV diagnostic tests utilized (child: 0–14 years old, young adult: 15–29 years, and elderly adult: 30 years). Before treatment, we collected sera from 104 males and 52 females. Sera from healthy subjects (controls) were also gathered from 68 males and 44 females (Table 2). We used 138 sera (76 from treated patients and 62 from untreated patients) to compare the humoral immune response to EBV DNase before and after therapy.

The qualitative evaluation of the response to Epstein-Barr virus (EBV) revealed the existence of specific IgG antibodies targeting the viral capsid antigen (VCA) in all of the patients studied, compared to 99.1% in the healthy subject group (Table 3). Furthermore, the patients tested positive for VCA IgA and EA IgG antibodies at rates of 89.7% and 95%, respectively. In contrast, these antibodies were found in just 42.8% and 47.3% of the healthy participants, respectively.

Interestingly, EA IgA antibodies were identified in more than two-thirds (100 out of 156) of the patients, while none of the control subjects showed evidence of these antibodies. Notably, the prevalence of anti-VCA and anti-EA IgA antibodies was significantly higher in the older patient cohort as compared to the child group (*p* = 0.00002 and *p* = 0.0031, respectively) and the young adult group (*p* = 0.004 and *p* = 0.029, respectively). In contrast, no significant changes were found between the groups of patients under the age of 30.

The occurrence of VCA-IgA was more frequent among young adult (aged 15–29 years) male patients compared to females (95.2% vs. 72.7%, respectively, with a *p*-value of 0.0097). There were no significant differences in any other markers between the male and female groups (Table 4).

When geometric mean antibody titers (GMT) were compared, there was a significant difference in VCA-IgG levels between patients and healthy people (4210.7 vs. 785.6). Patients had IgG antibody titers to EBV VCA that were equal to or more than 5120, but only 15.3% of healthy people had similar titers. In particular, some older patients aged 30 and up had increased IgA levels to VCA, exceeding 5120, when controls did not exceed 40. There was a difference in IgA anti-VCA levels between patients and controls (212 vs. 6.4).

IgA and IgG anti-EA titers in the patient group ranged from 10 to 1280 and 10 to 2560, respectively. A proportion of 3.2% of cases had very increased EA-IgG titers of 2560, whereas normal subjects did not exceed 80. GMT values for anti-EA-IgG and -IgA antibodies in patients were 196.2 and 27, respectively, and 10.6 and 5, respectively, in controls.

The differences in GMT values between patients and controls are further demonstrated in Figure 2, which shows the frequency of antibody titers against particular antigens. In terms of antibody titers, patients had higher percentages than controls for various antigens: elevated titers of VCA-IgG (5120) were in 61.5% of patients and 28.6% of controls, the mean antibody titer of VCA-IgA (640) was in 24.4% of patients and 8% of controls, EA-IgG (640) was in 16.7% of patients and 17.9% of controls, and low titers of EA-IgA (160) in 12.

All disparities noted between the patient and control groups exhibited highly significant statistical significance for antibody titers against both antigens (*p* < 10^−6^) (Table 5).

Geometric mean titers (GMT) were utilized to present the antibody titers for both VCA and EA markers. The statistical analysis involved calculating *p* values through the Wilcoxon–Mann–Whitney test.

When the IgG antibody response to VCA antigens was measured, the child patient group had titers ranging from 640 to 10,240, whereas young adults and adults had titers ranging from 1280 to 10,240. GMT values for VCA-IgG were 3377.9, 4399.5, and 4477.1, respectively, according to patient age groups. The GMT values for the control group, on the other hand, were significantly lower: 640, 1068.2, and 746.5, respectively. Titers ranged from 40 to 2560 for children, whereas they ranged from 160 to 2560 for young and elderly people. The elevated levels of VCA-IgG observed in children can be attributed to two factors: the difficulties involved in early illness detection due to the lack of identifiable clinical indications, and the rapid progression of the tumor seen in this age group. The corresponding distribution patterns of NPC cases in the youngest age group and cases of mononucleosis (IM) in Algeria are not a coincidence (article in process). IM is a self-limiting proliferation that is most likely linked to the activities of specific T cells. While its failure is infrequent, it can result in a malignant development similar to that found in NPC. Explaining why this natural occurrence does not occur in NPC would provide an important element of the solution.

In terms of VCA-IgA, the GMT values for the patient groups were 481, 170.7, and 625.8, respectively, with corresponding titer ranges of 40–640, 20–5120, and 40–10,240. In comparison, the GMT values for these antibodies in the control group were 8.3, 5.1, and 5.6. The patient group GMTs for anti-EA-IgG/IgA were 139.2, 174.4, and 261.6 for IgG and 13.1, 20, and 52.3 for IgA. These GMTs were 8.3, 7.6, and 24.2 for IgG in the control group. Particularly, EA-IgA titers in the child patient group did not reach 160, whereas 44.4% of sera from young adults had titers equal to or exceeding 320 (Figure 3).

When VCA-IgA and EA-IgA antibody titers were compared depending on patient age, there were significant differences between young adults and older individuals (*p* = 0.00013 and *p* = 0.0016, respectively). Furthermore, for VCA-IgA, significant differences were identified between the children’s group and other age categories (*p* = 6 × 10^−3^ and *p* = 10^−6^, respectively), as well as between children and older adults for EA-IgA (*p* = 0.0024). Anti-VCA and anti-EA-IgG antibody titers did not differ significantly among patient age groups, however.

The comparative analysis of VCA- and EA-IgA/IgG titers across age and gender in patients highlighted elevated GMTs of VCA- and EA-IgA/IgG in males in contrast to females. Among young patients, the GMTs for VCA-IgA were 253.9 in males and 80 in females. In patients aged 30 years and above, the GMTs for VCA-IgA were 706.6 in males and 485 in females, while for EA-IgG, they were 289.8 in males and 211.1 in females. A notable disparity based on age or gender distribution was evident in males compared to females for VCA IgA (*p* = 0.032).

The GMT distribution for VCA and EA antibody levels showed an increased trend in VCA-IgA and EA-IgA/IgG titers with illness development (Figure 4). For VCA-IgG, significant differences in GMT were seen between the pediatric group and the 15–29 year age range (*p* = 0.085). Similarly, significant differences in VCA-IgA and EA-IgG were found between the young adult group and those aged 30 years or older (*p* = 0.035 and *p* = 0.023, respectively).

Comparing antibody titers against VCA and EA for disease stages unveiled notable distinctions in anti-VCA and anti-EA IgG titers (*p* = 0.04 and *p* = 0.034, respectively). Given the relatively large number of patients under thirty and an even greater number of elderly patients who develop these antibodies, anti-VCA-IgA and anti-EA-IgG provide substantial arguments for tumor identification in this study. However, when employed on children and young patients, the utility of anti-EA IgA antibody testing appears to be diminished.

### 3.1. IgG Adsorption

Following the pre-absorption of IgG antibodies with the RF absorbent (Behring), there is no indication of competitive interaction between IgG and IgA in the patient’s serum. Post-treatment, these sera continued to test negative for VCA-IgA.

### 3.2. Assessment of Total Serum Immunoglobulins

All individuals’ total serum IgA levels were normal (Figure 5). Surprisingly, some people in the pediatric and young adult populations had undetectable blood VCA-IgA levels while maintaining normal (or increased) total IgA levels.

### 3.3. Antibody Reaction Patterns toward EBV DNase in Individuals with UCNT

The immune response of patients to recombinant EBV DNase was evaluated across age groups. The results were classified based on the anti-DNase IgA/IgG reaction, regardless of whether therapy had been given or not (Figure 6). To improve sensitivity, we used DNase produced from the P3HR-1 cell line, which demonstrated higher reactivity with sera in both immunoblot and ELISA experiments when compared to DNase obtained from the B95–8 strain [25].

**Figure 6 viruses-15-02158-f006:**
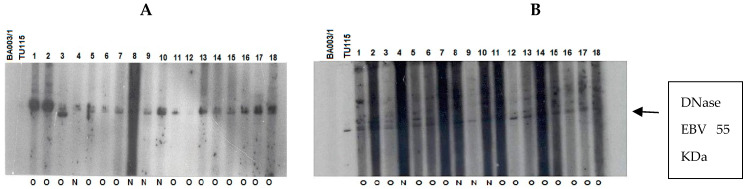
Immunoblot–ECL analysis of antibody responses to recombinant EBV DNase in patients with UCNT. The results obtained showed differences in the intensity of the IgA and IgG response between groups of patients who have undergone (O) or not (N) treatment. (**A**) IgA reactivity in children (1–17); (**B**) IgG reactivity in children (1–18); (**C**) IgA reactivity in young adults (1–19); (**D**) IgG reactivity in young adults (1–19); (**E**) IgA reactivity in adults (1–17); (**F**) IgG reactivity in adults (1–18). BA00/3 represents the absence of reactivity of the control serum of a European EBV-negative individual. TU115 represents the reactivity of the control serum of a Tunisian subject with EBV-positive NPC. EBV DNase of around 55 KDa is revealed in the margin of the ovalbumin (MW 46.103 kDa) and bovine albumin (MW: 66.103 kDa) markers. The intensity of DNase immunoreactivity was rated on a scale of 0 to 4 (0 for no staining, 1 for mild immunoreactivity, 2 for moderate immunoreactivity, 3 for high immunoreactivity, and 4 for very strong immunoreactivity) (Table 6).

Children and young adults developed IgA anti-DNase antibodies at higher levels than adults aged 30 and up. In contrast to children, young adults and adults had higher levels of IgG anti-DNase. For example, DNase reactivity with specific IgA antibodies was 4+ in 100% of instances in children before treatment. IgA anti-DNase antibodies were found in 34% of instances with a 4+ intensity and 52.1% of cases with an average intensity of 2+ in young adults. In comparison, 90.9% of elderly persons had a lower IgA anti-DNase reaction with a 1+ intensity. When treated participants were analyzed, this trend became much more pronounced.

Following treatment, 53.3% of children had IgA anti-DNase with a 3+ intensity, whereas 40% had a 4+ intensity. The majority (66%) of young adults had a 4+ intensity. Adults, on the other hand, displayed a 1+ intensity in 100% of cases. The explanation for the increased signal intensity following treatment in adolescents and young adults could be linked to the selection of post-treatment subjects based on recurrence, which could be associated with a high percentage of intense signal intensity.

Regarding IgG anti-DNase response intensity before treatment, children exhibited a 1+ intensity in 100% of cases. Young adults demonstrated a 3+ response intensity to IgG anti-DNase in 86.9% of cases. Among adults, only 9% exhibited a 4+ response intensity, while 90.9% exhibited a 3+ intensity. Remarkably, the children in our study only showed weak IgG responses.

Sera from six children (4+ intensity) and four adults (1+ intensity) were titrated to establish the reproducibility of the obtained data. The titer of IgA anti-DNase in children with a 4+ intensity exceeded 10,000, whereas it did not exceed 3000 in adults with a 1+ intensity. If this titration were performed on a larger number of sera, the increased IgA intensity in children would gain more credibility. Analyzing sera lacking IgA against VCA in both this study and a prior study [6] revealed that all patients had DNase-specific antibody response patterns that were closely related to age (Table 7).

## 4. Discussion

The prevalence of undifferentiated carcinoma of the nasopharynx (UCNT) cases in Algeria suggests the existence of at least three distinct age groups. This classification is critical for understanding anti-EBV antibody data. It is envisaged that patients will have IgG antibodies against VCA at an age when the majority of the community has had contact with the virus. Surprisingly, 48% of healthy people had IgG anti-VCA antibody titers comparable to the sick. As a result, these antibodies may only have limited diagnostic usefulness for NPC. In contrast, anti-EA IgG/IgA and anti-VCA IgA titers were significantly higher in healthy participants. Notably, GMTs of these antibodies in Algerian patients exceeded those observed in high-risk areas [26]. This divergence likely reflects the delayed consultation of patients in intermediate-risk regions. EBV serology exhibits characteristics closely correlated with undifferentiated and poorly differentiated non-keratinizing carcinoma of the nasopharynx. Recent studies have highlighted the significance and specificity of anti-EBV IgA antibodies in the prognosis and diagnosis of NPC [27,28]. Several risk factors for increased IgA anti-VCA antibodies have been highlighted [29,30,31]. VCA-IgA and EA-IgA antibody levels are a good diagnostic tool for patients aged 30 and up. It is critical, however, to note the absence of IgA anti-VCA in young patients. In our earlier study [6], this absence of IgA anti-VCA was identified in a significant proportion of patients under 30 years of age (50%) and a moderately high proportion of adult patients. This observation is consistent with the findings of North African authors [5,7]. Given the insufficient results in an intermediate-risk zone, this strategy had not been used in Algeria due to the resource-intensive nature of systematic NPC screening as conducted in China [32].

Since its beginning in 1966 by Henle and Henle [33], the traditional serological technique utilized in the investigation of disorders linked to EBV has been the subject of substantial research. Old’s original observation (in 1966) confirming the association of EBV with NPC was based on this serological foundation [34]. Subsequent research has shown that people with NPC have elevated antibody levels against EBV antigens, with a focus on IgA antibodies targeting VCA and EA antigens, which are important diagnostic indicators for NPC. These studies, however, have increasingly revealed limits in terms of their occurrence and specificity. Even though the results for the intensity of the IgA and IgG response to recombinant DNase were verified on a small number of sera, they appear interesting, particularly among children.

IgA anti-EA antibodies, which have been established as a particular marker for NPC [35], were found in 64.1% of patients in our investigation, but not in healthy people. Notably, no patient had a substantially greater titer of IgA anti-EA antibodies than IgA anti-VCA antibodies, which is consistent with previous findings [27,36]. Furthermore, EA-IgG antibodies were discovered to aid in NPC diagnosis, appearing in 96.7% of patients aged 30 years and older and 56.2% of cases among healthy people.

Except for VCA-IgA levels, which were greater in young Algerian patients, our findings on the frequency of antibodies against classical EBV antigens are similar to investigations by North African authors [37]. Patients’ higher levels of VCA-IgA, EA-IgG, and, to a lesser extent, EA-IgA are consistent with Tiwawech et al. [38]. Gurtsevitch et al. [39] discovered in non-endemic NPC locations that all cases of undifferentiated or poorly differentiated NPC have high amounts of IgA antibodies specific to conventional EBV markers. These higher antibody levels, however, have also been found in other head and neck illnesses [40].

Due to delayed patient consultations, the progression of VCA and EA antibody titers with the UCNT stage is difficult to evaluate. Our study found a significant age-related difference, with 4.2% of patients under 30 years old identified with stage II, 25.5% diagnosed with stage III, and 70.2% diagnosed with stage IV. Only 3.2% of patients aged 30 and up were diagnosed with stage I, 9.6% with stage II, 29% with stage III, and 58% with stage IV. In contrast, in industrialized nations, 11% of patients in Finland and 24% in Sweden are diagnosed with stage I [41,42]. It is critical to examine this data when comparing the performance of poor and developed countries, even though recruitment bias may contribute to these differences.

GMTs of anti-VCA and IgA anti-EA at the clinical stage revealed that the titers of these antibodies increased as the disease progressed. This finding is consistent with Tiwawech et al.’s observations [38]. Elevated VCA-IgA titers were especially noticeable in elderly persons with stage IV disease, and stage III patients had greater VCA-IgG titers than stage IV patients. However, no correlation was discovered between the clinical stage and GMT of VCA-IgG, which is consistent with the findings of Gurtsevitch et al. [39]. GMTs of VCA-IgG and VCA-IgA in stages I and II were higher than those in stages III and IV, according to the findings of these authors. When antibody titers were compared to evolutionary stages, substantial differences in IgG anti-VCA and IgG anti-EA were discovered in Algerian patients.

In Algeria, the distribution patterns of nasopharyngeal cancer (NPC) cases in the youngest age group and instances of infectious mononucleosis (IM) are not coincidental (article in process). IM is distinguished by self-limiting proliferation, which is frequently related to the function of specific T cells. While its malfunction is uncommon, it can result in a progression as extreme as that observed in NPC. Understanding why this natural phenomenon does not appear in NPC scenarios might provide a partial solution to this puzzle. The absence of IgA anti-VCA in patients under 30 years of age remains unexplained.

Although serum total IgA levels in these patients were normal or higher, no link was established between increased serum IgA levels and VCA IgA antibody titers. This observation is consistent with the findings of Abid et al. [43], but it varies from those of Wara et al. [44] and Baskies et al. [45], who found elevated serum IgA specific to EBV in NPC patients.

The observation of undetectable serum VCA-IgA levels in North African patients under thirty could have several potential explanations: (1) recent EBV infection in this region, where immune systems may not have had sufficient time to produce detectable levels of VCA-IgA antibodies; (2) variability in EBV strains circulating in North Africa among younger individuals that may not strongly induce VCA-IgA antibody production; or (3) individual genetic factors that influence an individual’s immune response to infections, including EBV.

The diagnostic performance of EBV antibody testing for NPC differs depending on the antigen and technology used [46,47]. Combining EBV antibodies against mixed antigens with other serologic markers has enhanced precision in NPC diagnosis or prognosis [48,49]. However, these tests are not without limitations in terms of reliability [50,51,52]. The use of viral load quantification alongside serologic tests seems to enhance sensitivity and specificity [53,54,55], although not all studies support this observation [56,57].

Technological advancements have resulted in the development of novel EBV markers, which may improve the diagnostic and prognostic accuracy of NPC. Several EBV genes, including BARF1, LMP1, BGLF4, BRLF1, BALF3, and BGLF5, have been linked to genomic instability and NPC formation [58,59]. EBV DNase has attracted interest as a marker in NPC diagnosis because of its identification in other diseases and its connection with anti-DNase IgA in NPC patients [60,61,62]. In our investigation, all patients had IgA and IgG antibodies against DNase, but normal people had just 2% IgA and 25% IgG positive. Specific DNase antibody reactivity varied by patient age, with children and individuals under 30 years old having higher IgA levels than older adults [63,64].

## 5. Conclusions

The findings of this study highlighted the categorization of UCNT patients into distinct age groups, providing critical context for understanding anti-EBV antibody responses. The incidence and intensity of various antibody responses against EBV antigens were investigated, and substantial differences were found across age groups and disease stages. These findings have far-reaching implications for UCNT diagnosis and prognosis, as well as our understanding of EBV-related malignancies.

Among the significant findings was the identification of IgA anti-DNase antibodies as a potentially more sensitive marker than IgA anti-VCA antibodies. Particularly, a large number of individuals had negative IgA anti-VCA results, emphasizing the need for more accurate diagnostic markers. IFA and ELISA both have shown their efficacy as primary approaches for detecting anti-VCA and anti-EA IgA antibodies in UCNT patients in this context. For instances with a low EBV burden or a lack of IgA against traditional viral antigens, the novel approach of detecting IgA anti-DNase represents an attractive option. This test has the potential to dramatically improve diagnositic accuracy, providing doctors with an extra essential tool in the evaluation of UCNT patients.

## Figures and Tables

**Figure 1 viruses-15-02158-f001:**
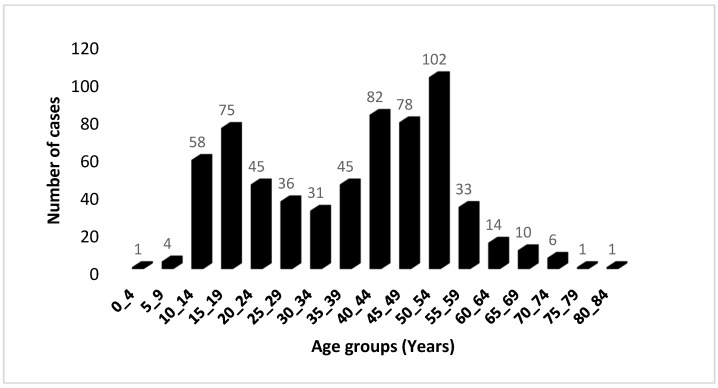
Age distribution of 622 UCNT cases in the Algerian population during the period 2012–2021. The distribution is classified by 5-year age groups.

**Figure 2 viruses-15-02158-f002:**
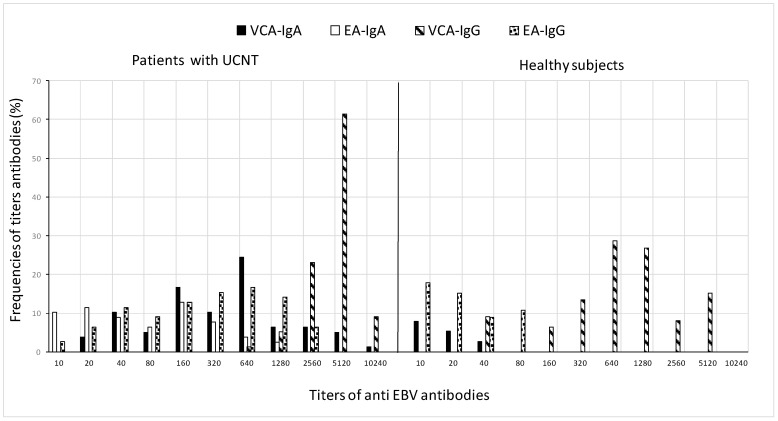
Frequency distribution of VCA and EA antibody titers in patients before treatment and in healthy subjects. For the calculation of GMT, all sera with a titer less than 10 in IFA were taken as 5, which is the reciprocal of the dilution.

**Figure 3 viruses-15-02158-f003:**
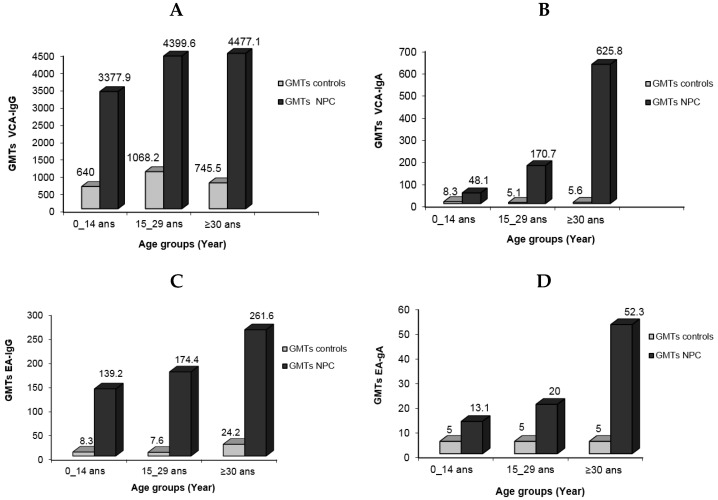
Geometric mean titers of EBV serological markers in NPC patients in different age groups compared to normal individuals: VCA-IgG (**A**), VCA-IgA (**B**), EA-IgG (**C**), and EA-IgA (**D**).

**Figure 4 viruses-15-02158-f004:**
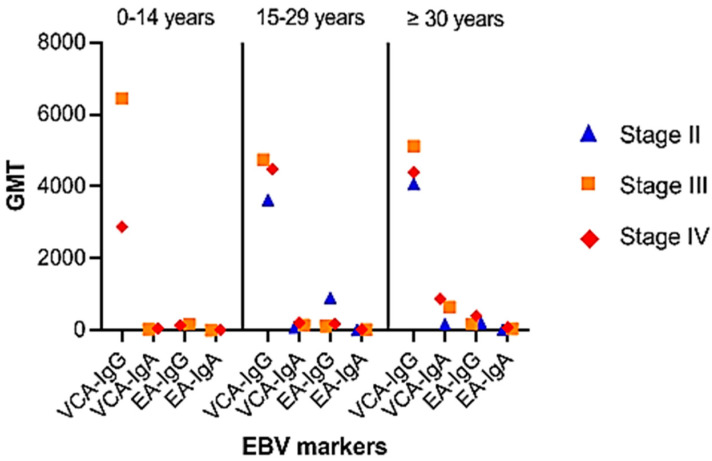
Distribution of patients by age group and disease stage: Group 0–14 years: stage III (n = 6), stage IV (n = 24). Group 15–29 years: stage II (n = 4), stage III (n = 18), stage IV (n = 42); group ≥ 30 years: stage I (n = 2), stage II (n = 6), stage III (n = 18), stage IV (n = 36). NE: not evaluable.

**Figure 5 viruses-15-02158-f005:**
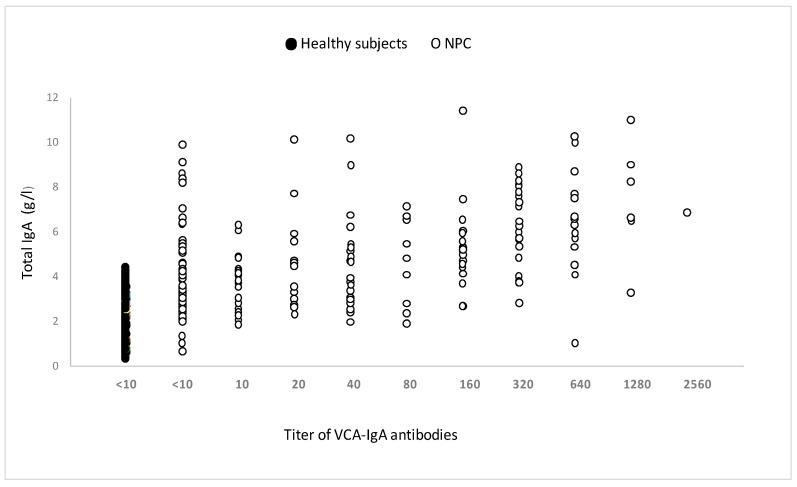
Total serum IgA in untreated patients with or without anti-VCA-IgA, measured by immunonephelometer. The measurement of total serum IgA was carried out in 84 patients responding or not to VCA-IgA and 82 healthy subjects of the same age.

**Table 1 viruses-15-02158-t001:** Distribution of patients with UCNT from the Otolaryngology Department of the Mustapha Pacha Hospital in Algiers, Algeria, and normal individuals by sex.

Sex	Patients		Healthy Subjects
	Untreated	Treated	
Males	104	41	68
Females	52	35	44
Total	156	76	112

**Table 2 viruses-15-02158-t002:** Distribution of untreated patients and normal individuals by age group and sex.

Age Group	Patients			Healthy Subjects		
	Number of Cases	Males	Females	Number of Cases	Males	Females
0–14 years	30	20	10	30	18	12
15–29 years	64	42	22	46	30	16
≥30 years	62	42	20	36	20	16
Total	156	104	52	112	68	44

**Table 3 viruses-15-02158-t003:** Prevalence of EBV serological markers in different groups of patients and normal subjects.

Subjects	NPC			Healthy		
Age Group	0–14 (n = 30)	15–29 (n = 64)	≥30 (n = 62)	0–14 (n = 30)	15–29 (n = 46)	≥30 (n = 36)
VCA-IgG	30 (100)	64 (100)	62 (100)	29 (96.7)	46 (100)	36 (100)
VCA-IgA	22 (73.3)	56 (87.5)	62 (100)	13 (43.3)	2 (4.3)	4 (11.1)
EA-IgG	28 (93.3)	60 (93.8)	60 (96.8)	19 (63.3)	16 (34.8)	28 (77.8)
EA-IgA	14 (46.7)	38 (59.4)	48 (77.4)	0 (0)	0 (0)	0 (0)

Numbers in brackets indicate the percentage of positive cases from the number of cases examined. n: number of subjects in each group.

**Table 4 viruses-15-02158-t004:** Prevalence of EBV serological markers in patients according to age and sex.

Antibodies	Sex	Total	Age (Years)		
			0–14	15–29	≥30
VCA-IgG	M	104 (66.7) *	20 (100)	42 (100)	42 (100)
	F	52 (33.3)	10 (100)	22 (100)	20 (100)
VCA-IgA	M	96 (68.6)	14 (70)	40 (95.2)	42 (100)
	F	44 (31.4)	8 (80)	16 (72.7)	20 (100)
EA-IgG	M	98 (66.2)	18 (90)	40 (95.2)	40 (95.2)
	F	50 (33.8)	10 (100)	20 (90.9)	20 (100)
EA-IgA	M	66 (66)	10 (50)	24 (57.1)	32 (76.1)
	F	34 (34)	4 (40)	14 (63.6)	16 (80)

* Numbers between brackets are positive proportions. ‘M’ indicates males and ‘F’ indicates females.

**Table 5 viruses-15-02158-t005:** Geometric mean titers of anti-EBV antibodies in patients compared to normal subjects.

Antibodies	GMT		*p* Values
	Patients	Healthy Subjects	
VCA-IgG	4210.7	785.6	*p* < 10^−6^
VCA-IgA	212	6.4	*p* < 10^−6^
EA-IgG	196.2	10.6	*p* < 10^−6^
EA-IgA	27	5	*p* < 10^−6^

**Table 6 viruses-15-02158-t006:** Reactivity of EBV anti-DNase antibodies in untreated and treated patients.

Age Group (Years)	Signal Intensity	Anti EBV DNase Antibodies (%) *			
		Before Treatment		After Treatment	
		IgA	IgG	IgA	IgG
Child (0–14) n = 21	4+	6 (100)		6 (40)	
	3+			8 (53.33)	
	2+				
	1+		6 (100)	1 (6.66)	15 (100)
	0				
Young adult (15–29) n = 65	4+	8 (34.78)		28 (66.66)	
	3+		20 (86.95)	6 (14.28)	26 (61.90)
	2+	12 (52.17)		4 (9.52)	
	1+	3 (13.04)	3 (13.04)	4 (9.52)	16 (38.09)
	0				
Adult (≥30) n = 52	4+	3 (9.09)	3 (9.09)		
	3+		30 (90.90)		19 (100)
	2+				
	1+	30 (90.09)		19 (100)	
	0				

* Percentage of DNase-positive subjects for each age group determined out of the total number of cases examined. Very high signal intensity was rated at 4+, high at 3+, medium at 2+, and low at 1+.

**Table 7 viruses-15-02158-t007:** Reactivity against EBV DNase in NPC patients without VCA IgA.

Age Group (Years)	Signal Intensity	Anti-DNase EBV	
		IgA	IgG
Child (0–14) n = 5	3+	5	
	1+		5
Young adult (15–29) n = 14	4+	4	
	3+		14
	2+	7	
	1+	3	
Adult (≥30) n = 12	3+		12
	1+	12	

## Data Availability

Data cannot be shared due to privacy restrictions.

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
