# Peer review of "A Distinct Anti-EBV DNase Profile in Patients with Undifferentiated Nasopharyngeal Carcinoma Compared to Classical Antigens"

_viruses, 2023, doi:10.3390/v15112158_

Round 1
Reviewer 1 Report
Major Comments
Although the authors recognized in their data presentation that Western Blotting is semi quantitative and gave relative value for band intensity, data evaluation could be greatly improved if the authors gave relative values to their reference standard TU115. This reviewer suggests giving TU115 a value of 1 and each sample lane could be scanned/pixel measured using freely available NIH Image-J software. After lane background subtraction a relative pixel value could be obtained using the equation : (NCP sample pixels – lane background pixels)/(TU115 pixels – background pixels) to obtain a semiquantitative Band value.
The authors may consider consolidating tables or place them in an appendix for minor observations (i.e. male vs female) and simply state in the results text the summarized findings
Minor comments
1 In the introduction text please add gene label BGLF5 and a corresponding reference ID for EBV DNAase to aid the reader.
2 The Figure discussed in the text appears to be out of synch with the actual Figure. It appeared results that referred to NPC staging, and anti-EBV IgG/IgA were stated as Figure 5 but in fact was Figure 4. The later figure text references appear also propagated the figure error. Please review the figure number listed in the text correspond to the correct figure.
It would aid the authors if the paper was re-edited for English grammar. This reviewer observed several instances of error in tense or syntax.
In general manuscripts are written in past tense not present
Author Response
Although the authors recognized in their data presentation that Western Blotting is semi quantitative and gave relative value for band intensity, data evaluation could be greatly improved if the authors gave relative values to their reference standard TU115. This reviewer suggests giving TU115 a value of 1 and each sample lane could be scanned/pixel measured using freely available NIH Image-J software. After lane background subtraction a relative pixel value could be obtained using the equation: (NCP sample pixels – lane background pixels)/(TU115 pixels – background pixels) to obtain a semi quantitative Band value.
Thank you for the reviewer's advice. Yes, we both believe that providing relative numbers for band intensity in contrast to a reference standard would be beneficial. Using the NIH Image-J software, we will rewrite our publication to include this analysis.
The authors may consider consolidating tables or place them in an appendix for minor observations (i.e. male vs female) and simply state in the results text the summarized findings
Dear Reviewer,
We appreciate your suggestion to simply state the summarized findings in the results text. We will delete the table (i.e. male vs female) hoping that this would be a good way to improve the readability and flow of our manuscript.
Thank you again for your valuable feedback.
Minor comments
1 In the introduction text please add gene label BGLF5 and a corresponding reference ID for EBV DNAase to aid the reader.
Thank you for your feedback. I have added the gene label BGLF5 and a corresponding reference ID for EBV DNAase to the introduction text:
2 The Figure discussed in the text appears to be out of synch with the actual Figure. It appeared results that referred to NPC staging, and anti-EBV IgG/IgA were stated as Figure 5 but in fact was Figure 4. The latter figure text references appear also propagated the figure error. Please review the figure number listed in the text correspond to the correct figure.
We have carefully investigated the concern you raised regarding the synchronization of the figures discussed in the text with the actual figures. Upon reevaluation, we have identified the source of the discrepancy and the corrections are highlighted in yellow.
Comments on the Quality of English Language
It would aid the authors if the paper was re-edited for English grammar. This reviewer observed several instances of error in tense or syntax.
We understand that there were instances of errors in tense or syntax in the manuscript. We sincerely apologize for any lapses in this regard. We will rectify this on the entire paper.
In general manuscripts are written in past tense not present
We appreciate your thorough review. We understand your concern regarding the use of verb tense. To address this, we will carefully review the entire manuscript and make the necessary adjustments to ensure that the appropriate past tense is consistently applied in the text.

Reviewer 2 Report
The authors compared EBV serological tests from three age groups in Algerian NPC patients, and observed the titers of IgA against VCA and EA were different in these three groups. Moreover, the authors indicated that the intensity of anti-DNase IgA may be a potential marker for detection of EBV-associated NPC. Some specific comments are provided below for the authors’ consideration.
The title of this manuscript is related to “A distinct anti-EBV DNase profile”, but most of the study (before page 11, line 286) discussed the classical EBV serological tests in different age groups. It would be beneficial to emphasize the main findings in this study. Additionally, the reasons for selecting EBV DNase as a test marker in NPCs should be further explained in the text (page 2).
Table 3. It is unclear why exactly 30 samples were chosen from each age group in these experiments. This number may need clarification or correction.
Page 12, lines 306-307. The authors quantified the intensity score of 0-4, so it would be helpful if they could provide more detailed descriptions of how this scoring system was applied. Furthermore, the authors only detected anti-EBV DNase in NPC cases (Table 7). It would be valuable to compare the titers of anti-EBV DNase between NPC patients and healthy individuals?
Table 7. The label of the children's group may be missing from this table.
Author Response
The authors compared EBV serological tests from three age groups in Algerian NPC patients, and observed the titers of IgA against VCA and EA were different in these three groups. Moreover, the authors indicated that the intensity of anti-DNase IgA may be a potential marker for detection of EBV-associated NPC. Some specific comments are provided below for the authors’ consideration.
The title of this manuscript is related to “A distinct anti-EBV DNase profile”, but most of the study (before page 11, line 286) discussed the classical EBV serological tests in different age groups. It would be beneficial to emphasize the main findings in this study.
The first section of the manuscript (before page 11, line 286) goes on traditional EBV serological tests in various age groups. This section includes crucial background information on EBV infection frequency and the limitations of existing serological assays.
Anti-VCA IgA and anti-EA IgG antibodies were found to be useful in diagnosing EBV tumors in this investigation, particularly in younger and older persons who are more likely to develop these antibodies. In contrast, anti-EA IgA antibodies appear to be less effective in identifying EBV malignancies in children and young adults.
Additionally, the reasons for selecting EBV DNase as a test marker in NPCs should be further explained in the text (page 2).
Dear Reviewer,
We agree that the reasons for selecting EBV DNase as a test marker in nasopharyngeal carcinoma (NPC) should be further explained in the text.
We have added a new paragraph to the introduction of our manuscript that explains the reasons for selecting EBV DNase as a test marker in NPC. We hope that this addition will make the manuscript more informative for readers.
Table 3. It is unclear why exactly 30 samples were chosen from each age group in these experiments. This number may need clarification or correction.
We understand your concern about the sample size of 30 for each age group in our experiments (table 3). By the way, we made transcription errors in the numbers. We added the right data in the table.
Page 12, lines 306-307. The authors quantified the intensity score of 0-4, so it would be helpful if they could provide more detailed descriptions of how this scoring system was applied.
Each sample lane was scanned and pixel measured using the software ImageJ (v. 1.8.0, Bethesda, MD, USA) for normalized quantification of the western blot
This was added in Methods.
Furthermore, the authors only detected anti-EBV DNase in NPC cases (Table 7). It would be valuable to compare the titers of anti-EBV DNase between NPC patients and healthy individuals?
It is interesting to examine the levels of anti-EBV DNase antibodies in NPC patients and healthy individuals, as this could provide insight into the link between EBV and NPC. But we did not do a comparison since IgA anti-DNase antibodies were found in just 2% of healthy patients.
We added a sentence in the bottom of the discussion.
Table 7. The label of the children's group may be missing from this table.
First, table 7 became table 6 in the revised manuscript.
Thank you for bringing this to my attention. I have carefully reviewed the table, and I acknowledge that the label for the children's group was inadvertently omitted. I will promptly add the appropriate label to ensure the clarity and completeness of the table.

Reviewer 3 Report
The authors recommend making drastic changes in their entire manuscript using the following comments and suggestions.
1. In the abstract lines 12-13: The NPC diagnosis (ADD MANILY) based on Epstein-Barr virus (EBV) detection in these three age groups has not been adequately evaluated in this region (WHICH YOU MAIN, ASIA OR Maghreb)?
2., the abstract clearly presents with corresponding values the classical serological test over the DNase test (the main target), please rephrase the entire abstract results to present the DNase test values similar to the classical one.
3. Lines 37-38: However, a notable population of North African patients under thirty exhibit undetectable serum VCA-IgA levels [5,6,7]. Is an excellent introduction part, but it needs to be completed by releasing an answer for why. The Professional’s reader worldwide may not know more about North Africa NPC
4. Lines 54-63: make the reader confused, are you speaking about the current results (By comparing these results to traditional …..) or previous results (so need refs), similarly, (The preliminary findings high) your results or published one (refs). If so, please transfer it into discussion or delete it.
5. From the introduction and Materials and methods it is hard to find: 1. If any previous work used anti-DNAse IgA/IgG antibodies titer in NPC (western blot)? 2. And subsequently, if YES from the author's lab or another's lab and where these are, please show. 3. If NO … so that the study title MUST BE changed to be VALIDATION …. 4. The authors are required to show us their patient samples from EBV causing NPC OR NON, also 5. With notification on patient’s family NPC history.
6. Please justify your entire study in light of the following studies: PMID: 21447725, 19336547, 28056971, 22095229, 21447725, 16106400, 15185352, 2161639.
Author Response
The authors recommend making drastic changes in their entire manuscript using the following comments and suggestions.
Dear Reviewer,
Thank you for your feedback on our manuscript. We appreciate you taking the time to provide us with such detailed and helpful comments and suggestions. We understand that you are recommending that we make drastic changes to our entire manuscript. We are committed to improving it, and we are open to making the changes that you have suggested.
- In the abstract lines 12-13: The NPC diagnosis (ADD MANILY) based on Epstein-Barr virus (EBV) detection in these three age groups has not been adequately evaluated in this region (WHICH YOU MAIN, ASIA OR Maghreb)?
We appreciate this insightful observation. Our study primarily focuses on the Maghreb region within North Africa. We acknowledge that the assessment of NPC diagnosis is mainly (word added in the text) based on EBV detection and the corrections were highlighted, in the revised manuscript, the specific geographical scope of our research.
2., the abstract clearly presents with corresponding values the classical serological test over the DNase test (the main target), please rephrase the entire abstract results to present the DNase test values similar to the classical one.
Thank you to the reviewer's request to rewrite the abstract results to present the DNase test values similarly to the classical method. We revised the abstract results to present the DNase test values in a format that aligns more closely with the classical method.
- Lines 37-38: However, a notable population of North African patients under thirty exhibit undetectable serum VCA-IgA levels [5,6,7]. Is an excellent introduction part, but it needs to be completed by releasing an answer for why. The Professional’s reader worldwide may not know more about North Africa NPC. In the manuscript text, we suggested explanations regarding the absence of anti-VCA IgA among particular NPC patients
The observation of undetectable serum VCA-IgA levels in North African patients under thirty could have several potential explanations: These are now mentioned in the manuscript.
- Lines 54-63: make the reader confused, are you speaking about the current results (By comparing these results to traditional …..) or previous results (so need refs), similarly, (The preliminary findings high) your results or published one (refs). If so, please transfer it into discussion or delete it.
Thank you for your suggestion. We will delete it from its location and move the content into the discussion section to provide a more appropriate context for the information.
- From the introduction and Materials and methods it is hard to find: 1. If any previous work used anti-DNAse IgA/IgG antibodies titer in NPC (western blot)? 2. And subsequently, if YES from the author's lab or another's lab and where these are, please show. 3. If NO … so that the study title MUST BE changed to be VALIDATION …. 4. The authors are required to show us their patient samples from EBV causing NPC OR NON, also 5. With notification on patient’s family NPC history.
Dear Reviewer,
We appreciate your careful review and your thoughtful comments.
Yes, there have been a few previous studies that have used anti-DNAse IgA/IgG antibodies titer in NPC with different assays (references 3, 15, 18, and 62 used in the present manuscript). Certain studies found that anti-DNAse IgA and IgG antibodies antibodies, but that they were less specific than anti-DNAse IgA antibodies. But our lab has not previously published any studies on this topic.
- Please justify your entire study in light of the following studies: PMID: 21447725 (Prognostic utility of anti-EBV antibody testing for defining NPC risk among individuals from high-risk NPC families. Kelly J Yu), 19336547 (Independent effect of EBV and cigarette smoking on nasopharyngeal carcinoma: a 20-year follow-up study on 9,622 males without family history in Taiwan. Wan-Lun Hsu.), 28056971 (Inhibition of Epstein-Barr virus reactivation by the flavonoid apigenin. Chung-Chun W), 22095229 (Correlates of anti-EBV EBNA1 IgA positivity among unaffected relatives from nasopharyngeal carcinoma multiplex families. C M Chang), 16106400 (Distribution of Epstein-Barr viral load in serum of individuals from nasopharyngeal carcinoma high-risk families in Taiwan. Xiaohong Yang), 15185352 (Epstein-Barr virus seroreactivity among unaffected individuals within high-risk nasopharyngeal carcinoma families in Taiwan. Amy Pickard), 2161639 (Multiple risk factors of nasopharyngeal carcinoma: Epstein-Barr virus, malarial infection, cigarette smoking and familial tendency. C J Chen).
Thank you for your comments on our draft. We appreciate your thorough review and insightful remarks. You have asked us to justify our entire work in light of the aboves researches. We are aware of various studies on the association between EBV infection and NPC. However, we believe that our study contributes to the field in a unique way by particularly evaluating the potential of anti-DNAse IgA/IgG antibody titer as a diagnostic marker for NPC. This is significant, especially given the unique epidemiology of UCNT in the Maghreb, which affects all age groups.
We feel that our research has the potential to improve NPC identification and diagnosis. We expect that by developing a more specific and sensitive test marker for NPC, we will be able to help more patients receive the necessary early diagnosis and treatment.
We appreciate your consideration of our study.
Sincerely,

Reviewer 4 Report
Melouli et al. submitted the manuscript titled: "A distinct anti-EBV DNase profile in patients with undifferentiated nasopharyngeal carcinoma compared to classical antigens".
The major goal of submitted study is to assess EBV DNase reactivity in Algerian UCNT patients, with a focus on those who do not have detectable serum IgA responses against VCAs and are divided into three age groups.
The manuscript is nicely written. Some English editing is needed. The number of patients for the study is rather small, however, due to the specificity of the disease can be taken into consideration. The performed statistical analysis is sufficient, however some of the methods could be improved (e.g. Immunonephelometry).
The results are in the line with the references/literature, and well described.
Authors describe that When the IgG antibody response to VCA antigens was measured, the child patient group had titers ranging from 640 to 10240, whereas young adults and adults had titers ranging from 1280 to 10240. It would be nice to write a small paragraph especially for child age group what factors influence this range in terms of: is it genetic predisposition, environmental factors etc. How this influences the risk of NPC development especially in young children?
Also, what are the limitations of this study? This should be included in conclusion section.
Author Response
Authors describe that When the IgG antibody response to VCA antigens was measured, the child patient group had titers ranging from 640 to 10240, whereas young adults and adults had titers ranging from 1280 to 10240. It would be nice to write a small paragraph especially for child age group what factors influence this range in terms of: is it genetic predisposition, environmental factors etc. How this influences the risk of NPC development especially in young children?
Dear Reviewer,
Thank you for your feedback on our manuscript. We appreciate your careful review and your thoughtful comments.
We added a paragraph to the discussion part of our publication that specifically highlights the factors that determine the range of IgG antibody response to VCA antigens in children.
It should be noted, however, that the relationship between the range of antibody response to VCA antigens and the risk of NPC development in children is complicated and not entirely understood. More investigation is required to identify the precise nature of this link.
Sincerely.
Also, what are the limitations of this study? This should be included in conclusion section.
Limitations of the research are added in conclusion section.
There are various drawbacks to this study. The most important, the sample size is modest in children patient group. As a result, the generalizability of our findings to a larger population is limited.
Comments on the Quality of English Language

Round 2
Reviewer 2 Report
Thanks for the authors’ careful revisions, which have resulted in improvements to the manuscript based on the suggestions. However, several specific comments are mentioned below for the authors’ consideration.
Table 3. In the revised version, did they choose the same samples (30, 64, and 62) for both NPC and Healthy subjects in the experiments? This is not convincing me for the revised manuscript.
Table 6. Although the authors explained how they quantified the intensity score based on signal intensity, I’m still confused about how they define the scores as mild, moderate, high, or very strong immunoreactivity (lines 322-325). What’s the direct relationship between “RPV (relative pixel value)” and the signal intensity or the intensity score? This could significantly impact the final conclusions.
Reviewer 3 Report
Thank you for your reply.